# HueManity: Probing Fine-Grained Visual Perception in MLLMs

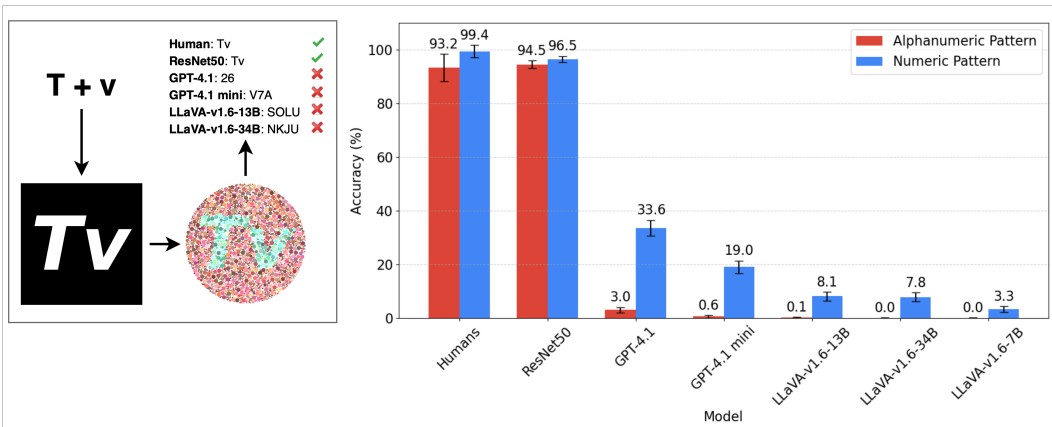

Figure 1: We present **HueManity** - an automated scalable benchmark for evaluation of fine-grained visual perception in MLLMs. The pipeline (left) embeds characters within challenging Ishihara-style image patterns, while ensuring human readability of the generated images. Experiments reveal (right) that humans and a fine-tuned ResNet50 baseline significantly outperform top-5 leading closed-source and open-source MLLMs, exposing a critical lack of fine-grained understanding.

## Abstract

Recent Multimodal Large Language Models (MLLMs) demonstrate strong high-level visual reasoning on tasks such as visual question answering and image captioning. Yet existing benchmarks largely overlook their ability to capture fine-grained perceptual details. As MLLMs are increasingly deployed in safety and reliability critical settings, perceptual acuity becomes essential. We present Hue-Manity, a scalable automated benchmark for assessing fine-grained visual perception in MLLMs. HueManity comprises 83,850 Ishihara-style images embedding alphanumeric strings, designed to evaluate pattern recognition, a core aspect of visual understanding. Our evaluation of nine state-of-the-art MLLMs uncovers a striking performance deficit: the strongest model achieved only 33.6% accuracy on a simple numeric task and 3% on a harder alphanumeric task, compared to near-ceiling performance from humans (99.38%, 93.25%) and a fine-tuned ResNet-50 (96.5%, 94.5%). These findings expose a critical weakness in MLLMs' perceptual grounding, one that remains obscured by conventional benchmarks emphasizing high-level semantics.

## 1 Introduction

The trajectory of Multimodal Large Language Models (MLLMs) Comanici et al. (2025); Achiam et al. (2023); Bai et al. (2023a); Li et al. (2023b); Gong et al. (2023); Liu et al. (2024a; 2023); Anthropic (2025) has been marked by impressive advancements, demonstrating sophisticated capabilities in bridging visual and textual information. Their skills extend beyond simple image labeling Russakovsky et al. (2015); Deng (2012), enabling complex tasks like generating detailed image descriptions Dong et al. (2024); Fu et al. (2024b), answering intricate visual questions requiring

inference about relationships and activities Weng et al. (2025); Chen et al. (2025); Kuang et al. (2024), and participating in nuanced dialogue about visual content Cao et al. (2024). This success is largely attributed to pre-training on vast, web-scale image-text datasets, which has cultivated a powerful ability to map high-level semantic concepts between modalities Jia et al. (2021); Radford et al. (2021); Schuhmann et al. (2022); Alayrac et al. (2022); Qi et al. (2020); Zhai et al. (2022); Pham et al. (2023).

However, this focus on semantic understanding, both in training and in predominant evaluation paradigms Bai et al. (2023b); Li et al. (2024; 2023a); Xu et al. (2024); Yin et al. (2023); Liu et al. (2023), has created a critical blind spot: the models' foundational perceptual acuity remains largely unprobed. Human vision is not merely a semantic engine, it is fundamentally a system for perceptual organization, excelling at extracting coherent signals from visually cluttered environments. This ability to discern patterns from subtle, low-level cues — like color, luminance, and texture — is a prerequisite for virtually all higher-level visual reasoning. This is a skill honed by millennia of evolution; before our ancestors could reason 'that is a tiger,' they first had to perceptually distinguish the subtle pattern of its stripes from the chaotic background of jungle leaves.

Existing benchmarks have predominantly centered on the conceptual capabilities of MLLMs, leaving their resilience to perceptual challenges like pattern recognition and feature differentiation in cluttered scenes largely unevaluated. This paper introduces **HueManity**, a benchmark specifically designed to probe this gap. Our methodology is inspired by Ishihara plates Clark (1924), a classic tool from human ophthalmology designed to isolate a specific perceptual skill: figure-ground segregation based on subtle color cues. It is crucial to clarify that HueManity does not aim to diagnose 'color blindness' in MLLMs. Instead, our Ishihara-style stimuli, created using controlled generation techniques, test an MLLM's fundamental ability to identify embedded alphanumeric characters by their subtle color and luminance contrasts within visually cluttered dot patterns.

The ability to parse characters from our Ishihara-style plates is a direct proxy for an MLLM's capacity to handle real-world visual challenges characterized by clutter, partial occlusions, and variable lighting. An MLLM that cannot distinguish a number embedded in a field of dots may likewise struggle to perform optical character recognition (OCR) on text seen through a rainy window or accurately transcribe text from a crumpled, faded receipt. Therefore, HueManity serves not merely as a test of pattern recognition, but as a diagnostic tool to probe the limitations that prevent MLLMs from achieving robust, human-like visual intelligence. Our findings reveal that this novel evaluation exposes a profound deficit in current models, a result with significant implications for deploying MLLMs in domains requiring human-like perceptual understanding, from autonomous systems to medical imaging Yang et al. (2025).

To address this identified gap and facilitate further research in this domain, this paper makes the following contributions:

1. **We introduce HueManity, a new large-scale benchmark (83,850 images)** featuring Ishihara-inspired alphanumeric stimuli. The benchmark utilizes a principled design with 25 carefully curated color pairs, selected using CIEDE2000 ($\Delta E_{2000}$) metrics and manual verification, ensuring both systematic challenge and fairness for human comparison.

2. **We conduct a comprehensive evaluation of nine state-of-the-art MLLMs**, revealing a significant performance gap when compared to strong human and fine-tuned ResNet50 baselines. This suggests MLLM limitations are architectural rather than the task being intractable

3. **We release open-source code for generating challenging Ishihara-style perceptual stimuli**, enabling reproducible research and community-driven extensions.

## 2 RELATED WORKS

### 2.1 MULTIMODAL MODELS

With the remarkable advancements of Large Language Models (LLMs), recent research has extended their capabilities to multimodal domains by integrating visual information, giving rise to Multimodal Large Language Models (MLLMs) Comanici et al. (2025); Achiam et al. (2023); Bai

et al. (2023a); Li et al. (2023b); Gong et al. (2023); Liu et al. (2024a; 2023). These models typically align visual features from pre-trained image encoders with LLMs via modality adaptation layers. Early works like BLIP-2 Li et al. (2023b) pioneered this architecture by first pre-training on image-text datasets and fine-tuning on task-specific benchmarks such as Visual Question Answering (VQA). Subsequent models like LLaVA Liu et al. (2023) advanced this approach by leveraging synthetic instruction-following data in VQA formats, significantly improving instruction tuning performance. More recent efforts have expanded into video understanding and even image generation Baldridge et al. (2024); Saharia et al. (2022), showcasing the versatility of MLLMs across modalities. However, this celebrated success in visual tasks often appears reliant on their powerful language capabilities for reasoning and interpretation, potentially overshadowing the need to scrutinize their fundamental perception skills. Addressing this gap, HueManity is a benchmark designed specifically to isolate and probe these visual abilities.

Table 1: Comparison of MLLM evaluation benchmarks across key methodological attributes. Novelty refers to the use of non-internet, generated images.

| Benchmark | Data Size | Automatic Annotation | Novel Images | Answer Form | Automated Evaluation |
|---|---|---|---|---|---|
| LVLM-eHub | - | ✓ | ✗ | Free-form | ✗ |
| MMBench | 3,217 | ✗ | ✗ | Multi-choice | ✓ |
| SEED-Bench2 | 24,371 | ✗ | ✗ | Multi-choice | ✓ |
| Blink | 3,807 | ✗ | ✗ | Multi-choice | ✓ |
| ZeroBench | 100 | ✗ | ✓ | Free-form | ✓ |
| **HueManity** | 83,850 | ✓ | ✓ | Exact Match | ✓ |

## 2.2 MLLM EVALUATION

While Multimodal Large Language Models (MLLMs) demonstrate strong capabilities in global image understanding, they often falter on fine-grained visual tasks requiring precise recognition and localization Huang & Zhang (2024); Li et al. (2024). In response, a variety of benchmarks have been developed to probe these limitations. However, many prominent benchmarks face significant methodological challenges related to scalability and subjectivity.

A primary challenge is the reliance on subjective and costly evaluation methods. For instance, benchmarks like LLaVA-Bench Liu et al. (2023), LAMM Yin et al. (2023), and TouchStone Bai et al. (2023b) employ GPT-based models as judges to assess the relevance and accuracy of model outputs, especially when, we show in this work that GPT-based models are unreliable for fine-grained perception. This approach introduces potential biases and reliability issues inherent in using another model as a reference. Similarly, LVLM-eHub Xu et al. (2024) aggregates multiple vision benchmarks but depends on expensive and subjective human annotators to compare model outputs.

To address these issues, other benchmarks adopted more objective, often multiple-choice formats. MME Fu et al. (2024a) pioneered this with Yes/No questions, while MMBench Liu et al. (2024b), MM-Vet Yu et al. (2023), SEED-Bench Li et al. (2024; 2023a), BLINK Fu et al. (2024c) expanded this approach to cover a wider range of sub-tasks like OCR and recognition. Despite their objectivity, these benchmarks often draw heavily from existing VQA datasets or use GPT-generated questions, raising concerns about data contamination and novelty. Furthermore, the tasks can lack sufficient difficulty, as evidenced by most open-source models achieving 40–60% accuracy. While the recently proposed ZeroBench Roberts et al. (2025) offers a difficult benchmark for image understanding using Gemini as a judge, its utility for assessing fine-grained perception is constrained by its very small scale and broader focus.

Another body of research leverages visual illusions to investigate the perceptual differences between humans and MLLMs, as seen in benchmarks like HallusionBench Guan et al. (2024), IllusionVQA Shahgir et al. (2024), and IllusionBench Zhang et al. (2025). Although valuable for revealing these differences, these benchmarks suffer from two primary limitations. First, they are often not scalable due to small dataset sizes. Second, their reliance on common internet images, potentially allowing MLLMs to perform well on images they have already seen during training.

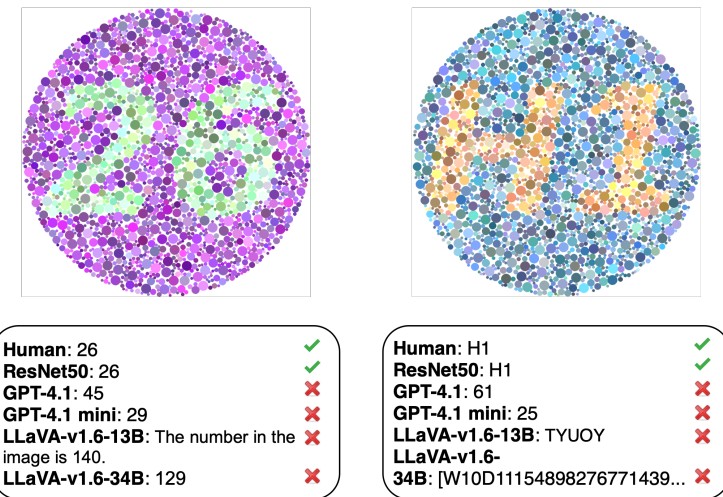

Figure 2: Qualitative examples showing predictions of 4 representative MLLMs vs baselines on numeric and alphanumeric tasks.

In contrast to these approaches, HueManity isolates a foundational visual skill: discerning patterns from subtle cues in cluttered environments. By using procedurally generated, Ishihara-style dot patterns with embedded alphanumeric characters, our benchmark offers three key advantages. First, procedural generation ensures data novelty and scalability, mitigating data contamination risks. Second, an exact-match evaluation framework provides a completely objective and automated scoring method, avoiding the biases of human or LLM-based judges. Finally, by controlling variables like color, shape, distribution, and sizes, HueManity offers a targeted and rigorous methodology for evaluating a core perceptual capability that remains a significant challenge for current MLLMs.

## 3 DATA CREATION

The HueManity benchmark is built upon a dataset of 83,850 images, each featuring a two-character alphanumeric string rendered as an Ishihara-style dot pattern, along with its corresponding ground truth label and generation parameters. Our generation process involves creating a base text mask, selecting a color palette, and rendering the final dot pattern. Figure 1 illustrates the core stages of this pipeline.

**Text Mask Generator.** The first step is to create a 900x900 pixel binary mask for each two-character string. We use the Pygame library to render the string in white on a black background. To ensure the characters are thick enough for dot-based rendering and remain clearly legible, we use the DejaVu Sans font, styled in bold and italic at a size of 550 (Figure 1).

**Color Pairs Selection** To generate the patterns, we meticulously selected 25 distinct foreground-background color pairs. This selection process involved a multi-stage procedure combining quantitative analysis using the CIEDE2000 color-difference formula Luo et al. (2001) with extensive manual verification. This ensures that the color pairs are balanced for perceptual difficulty while remaining legible to humans (see Appendix D for details).

**Ishihara-Style Pattern Generation.** Our pattern generator, adapted from an open-source Pygame project[1], iteratively populates the image with non-overlapping circles. Over 30,000 iterations, the generator randomly places circles, computes their maximum non-colliding radius (4-15 pixels), and assigns it a color. The color assignment depends on whether the circle's center falls within the character region of the text mask. This initial color then undergoes three randomized transformations: a gradient shift towards the other color, an RGB color shift (range [-30, +30]), and an RGB lightness scaling (factor 0.66 - 1.5). These transformed circles are then rendered, resulting in the final dense Ishihara-style pattern (Figure 2).

---

[1] https://github.com/hakrackete/Ishihara-color-plate-generator

## 4 EXPERIMENTAL SETUP

**Benchmark Tasks and Evaluation Sets.** From the generated stimuli, we designed a benchmark with two recognition tasks of varying difficulty and two visual conditions to isolate specific model capabilities.

There are two distinct recognition tasks of varying difficulty: a numeric task with a label space of 90 possible outputs and a more challenging alphanumeric task with 3,364 possible outputs (Refer to Appendix B.1). The larger search space makes the alphanumeric task inherently more difficult.

- *Number Recognition Set (Easier Task)*: This subset contains images with two-digit numeric strings (e.g., 17, 83, 65). We exclude leading-zero numbers ('00'-'09') to prevent ambiguity between single and double-digit interpretations.

- *Alphanumeric Recognition Set (Harder Task)*: This subset includes two-character strings (e.g., A7, 9b, XG) formed from lowercase letters (a-z), uppercase letters (A-Z), and digits (0-9). We excluded visually ambiguous characters ('l', 'I', 'J', 'O') from the character space.

Further, to disentangle the challenge of perceptual grouping from fundamental character recognition, we evaluate models under two visual conditions for each image.

- *Ishihara Pattern:* The primary task using the Ishihara-style dot pattern images.

- *Text Masks:* A control task using the binary text mask used in the creation of Ishihara Pattern images. This condition establishes a baseline for each model's fundamental OCR capability, helping to disentangle it from performance on the more perceptually complex patterns.

Due to computational and API cost constraints, our MLLM evaluations were conducted on subsets of 1,000 images, randomly sampled for both the numeric and alphanumeric tasks.

**Multimodal Large Language Models (MLLMs).** We evaluated a diverse set of nine Multimodal Large Language Models (MLLMs), including both commercial API-based and publicly available open-source models. Model inference was orchestrated using Promptfoo[2], a platform facilitating reproducible benchmarking through flexible prompt definition and API integration. Open-source models were hosted locally via Ollama[3] and inferred on a single NVIDIA A100 GPU. Images were Base64 encoded at 900x900 original resolution and submitted with the following task-specific prompts, which were kept consistent across all models and conditions:

> **Number Recognition Prompt:** *"What is the number in this image? Strictly stick to the format: Answer: [number in the image]"*
>
> **Text Recognition Prompt:** *"What is the exact text in this image? It has only alpha-numeric characters excluding small l, capital O, capital I, and capital J to avoid ambiguity. Strictly stick to the format: Answer: [exact text in the image]"*

**Human Performance Baseline.** To establish a human performance baseline, we recruited 16 adult volunteers from diverse age groups with self-reported normal color vision. Each participant was evaluated on 100 Ishihara pattern images for each of the tasks (numerical and alphanumeric recognition). These images were randomly sampled from the 1,000-image sets used in the MLLM evaluations. Participants viewed the 900x900 pixel images in a Google Sheets document and provided their responses to the same prompts given to the models, ensuring a direct and methodologically consistent comparison.

**Traditional Computer Vision Baseline (ResNet50).** As a representative traditional computer vision baseline, we trained and evaluated a ResNet50 model. We utilized a ResNet50 pre-trained on ImageNet, obtained from the PyTorch `vision` library. The standard classification layer of the

---

[2] https://www.promptfoo.dev
[3] https://ollama.com

ResNet50 was replaced with two independent classification heads. Each head was designed to predict one character, treating the task as two independent character recognition problems. For the purpose of fine-tuning this model, we utilized 2,000 images randomly sampled from the broader HueManity dataset, ensuring these were distinct from the final evaluation subsets. Training was conducted for 30 epochs using the Adam optimizer with a learning rate of $1e - 3$. The loss function was the sum of the cross-entropy losses calculated independently for each of the two classification heads. The trained model was evaluated on the same 1,000-image subsets used for the MLLM evaluations.

## 5 RESULTS AND ANALYSIS

### 5.1 HUMAN AND RESNET50 BASELINE PERFORMANCE

To contextualize the performance of MLLMs, we established two critical baselines: human evaluators and a fine-tuned ResNet50 model. These baselines serve to confirm the task's solvability and provide a clear performance ceiling.

Human evaluators demonstrated near-perfect accuracy, achieving 99.37% on the numerical task and 93.25% on the alphanumeric task. Annotator feedback indicated that the few errors on the alphanumeric task were due to confusion between visually similar characters (e.g., 's' vs. 'S', 'c' vs. 'C', 'w' vs. 'W'), not an inability to perceive the patterns. Annotators also report that the character recognition time was in the order of a few seconds. This near-flawless performance confirms that the stimuli are clear and the task is fundamentally solvable for a proficient visual system.

Similarly, a fine-tuned ResNet50 He et al. (2015) model achieved high accuracy, scoring 96.5% on the numerical task and 94.5% on the alphanumeric task. This strong performance from a standard convolutional architecture, trained on only 2,000 examples, proves that the perceptual cues within the images are sufficient for established computer vision techniques to learn the task.

Together, these baselines establish that the HueManity benchmark is a well-posed and learnable perception challenge, suggesting that any failures by MLLMs are likely due to architectural or training-related limitations rather than the inherent difficulty of the task itself.

Table 2: **Accuracy on the number and alphanumeric recognition tasks** for human evaluators, ResNet50, and various MLLMs on both text masks and patterned HueManity images. $\pm$ denotes the Wilson Confidence intervals at 95% confidence.

| | Number Task | | Alphanumeric Task | |
|---|---|---|---|---|
| | **Mask** | **Pattern** | **Mask** | **Pattern** |
| Random choice | 1.11% | 1.11% | 0.029% | 0.029% |
| Humans (average) | - | $99.37 \pm 2.37\%$ | - | $93.25 \pm 5.08\%$ |
| ResNet50 | - | $96.5 \pm 1.15\%$ | - | $94.5 \pm 1.42\%$ |
| API-based models | | | | |
| GPT-4.1 | $100 \pm 0.19\%$ | $33.6 \pm 2.92\%$ | $80 \pm 2.47\%$ | $3.0 \pm 1.07\%$ |
| GPT-4.1 mini | $100 \pm 0.19\%$ | $19.0 \pm 2.42\%$ | $72.4 \pm 2.76\%$ | $0.6 \pm 0.51\%$ |
| Claude 3.7 Sonnet | $100 \pm 0.19\%$ | $0.4 \pm 0.43\%$ | $82.2 \pm 2.36\%$ | $0 \pm 0.19\%$ |
| Open-source models | | | | |
| LLaVA-v1.6-34B | $96.6 \pm 1.13\%$ | $7.8 \pm 1.66\%$ | $27.1 \pm 2.75\%$ | $0 \pm 0.19\%$ |
| LLaVA-v1.6-13B | $87.2 \pm 2.07\%$ | $8.1 \pm 1.69\%$ | $31.8 \pm 2.88\%$ | $0.1 \pm 0.27\%$ |
| LLaVA-v1.6-7B | $87.7 \pm 2.03\%$ | $3.3 \pm 1.11\%$ | $15 \pm 2.21\%$ | $0 \pm 0.19\%$ |
| Mistral-small3.1-24b | $100 \pm 0.19\%$ | $0.1 \pm 0.27\%$ | $58.7 \pm 3.04\%$ | $0 \pm 0.19\%$ |
| Qwen VL Max | $100 \pm 0.19\%$ | $0.2 \pm 0.33\%$ | $83.5 \pm 2.29\%$ | $0 \pm 0.19\%$ |
| Pixtral | $100 \pm 0.19\%$ | $1 \pm 0.64\%$ | $65.8 \pm 2.93\%$ | $1.8 \pm 0.84\%$ |

## 5.2 MLLM Performance

The performance of the nine evaluated MLLMs on HueManity sharply contrasts with the near-perfect accuracies of both humans and the ResNet50 baseline (Table 2). All MLLMs consistently struggled, with the best achieving only 33.6% on the numeric task and a mere 3% on the alphanumeric task.

As expected, most models demonstrated strong Optical Character Recognition (OCR) capabilities on the simple binary mask images. For the numeric mask task, nearly all models achieved perfect or near-perfect accuracy. Performance on the more complex alphanumeric mask was more varied, while API-based models like GPT-4.1 and Claude 3.7 Sonnet maintained high accuracy (>80%), the open-source LLaVA family notably struggled (15-32% accuracy), indicating a weaker baseline OCR capability for complex character sets.

The models' high OCR accuracy on masks makes their widespread failure on the patterned images even more striking. This drop in performance pinpoints the challenge as a perceptual failure, not a simple recognition one. On the "easy" numeric pattern task, a clear performance hierarchy emerged. GPT-4.1 was the only model with meaningful success at 33.6% accuracy. A second tier, including GPT-4.1 mini (19.0%) and the LLaVA family (3.3-8.1%), showed minimal capability. The remaining models, such as Claude 3.7 Sonnet and Qwen VL Max, failed almost completely, scoring near zero. On the "hard" alphanumeric pattern task, MLLM performance collapsed almost entirely. GPT-4.1 was again the top performer but with a mere 3.0% accuracy. It was followed by Pixtral at 1.8%. All other evaluated models scored less than 1%, demonstrating a near-total inability to perform perceptual grouping on this more complex task.

Several characteristics inherent to the current design and training paradigms of many MLLMs may contribute to their observed difficulties on tasks demanding nuanced visual perception.

**Semantic Optimization in Pre-Training.** MLLM vision encoders are typically optimized to capture high-level semantic information for tasks like scene understanding Liu et al. (2023; 2024a); Agrawal et al. (2024); Bai et al. (2023a; 2025). This focus on global context may cause the loss of fine-grained local details, such as the subtle color and texture cues that define the characters in HueManity. Furthermore, pre-training on web-scale datasets of images paired with descriptive text may not provide sufficient exposure to stimuli requiring intensive perceptual organization without strong semantic anchors.

**Architectural Bottlenecks.** The architectures themselves may contribute to this failure. Many MLLMs use Vision Transformers (ViTs) Dosovitskiy et al. (2021) that divide images into patches, a process that can disrupt fine-grained details within those patches. Additionally, the projection layers that connect the vision encoder to the language model can act as an information bottleneck, abstracting away precise, high-resolution feature distinctions that are essential for solving our task.

## 5.3 Can MLLMs Learn the Task? Probing with In-Context Learning and Fine-Tuning

**In-Context Learning** We tested whether providing in-context examples could improve performance. While this approach retained 100% accuracy for the simple "mask" images, it proved ineffective and often detrimental for the patterned Ishihara plates (Table 3). For most models, performance degraded as more examples were added. These results are consistent with Shahgir et al. (2024)'s findings and we hypothesize that the visual complexity of the examples introduces noise that confuses the models, reinforcing the conclusion that the deficit is perceptual rather than a lack of contextual understanding.

**MLLM Fine-tuning** To determine if direct training could overcome the observed perceptual challenges, we LoRA fine-tuned the Gemma-3-4B model on 500 examples from the HueManity dataset using the HuggingFace SFTTrainer[4] with Adam optimizer, 2e-4 learning rate, and 3 epochs. The results were striking and revealed a critical limitation. While fine-tuning on the simple text masks improved performance on that control task as expected, it provided no benefit for recognizing the patterned Ishihara images. More critically, fine-tuning on the patterned images not only failed to improve performance on the core task but also destroyed the model's ability to solve the simple mask

---

[4]https://ai.google.dev/gemma/docs/core/huggingface_text_full_finetune

Table 3: Few-shot performance on Ishihara Pattern for number task with 95% Wilson confidence intervals.

| Model | 0-shot (Baseline) | 1-shot | 3-shot | 5-shot |
|---|---|---|---|---|
| GPT 4.1 | $33.6 \pm 2.9\%$ | $34.0 \pm 2.9\%$ | $35.0 \pm 3.0\%$ | $31.0 \pm 2.9\%$ |
| GPT-4.1 mini | $19.0 \pm 2.5\%$ | $14.0 \pm 2.2\%$ | $9.0 \pm 1.8\%$ | $6.0 \pm 1.5\%$ |
| Pixtral Large | $1.0 \pm 0.6\%$ | $1.0 \pm 0.6\%$ | $0.0 \pm 0.2\%$ | $0.0 \pm 0.2\%$ |
| Claude 3.7 Sonnet | $0.4 \pm 0.4\%$ | $0.0 \pm 0.2\%$ | $1.0 \pm 0.6\%$ | $1.0 \pm 0.6\%$ |
| Qwen 2.5 VL Max | $0.2 \pm 0.3\%$ | $0.0 \pm 0.2\%$ | $0.0 \pm 0.2\%$ | $0.0 \pm 0.2\%$ |

task, causing its accuracy to plummet from 80% to 0%. Instead of learning to perceive the characters, the model simply learned to output plausible-looking two-character strings (e.g., "4G", "6t") regardless of the visual input. This suggests the failure is rooted in deeper architectural limitations, as the model appears incapable of learning the necessary perceptual skill.

Table 4: Performance of fine-tuned Gemma3-4B and qualitative examples of predictions.

| Training Data | Task | Accuracy | Predictions for Ground Truth Values | | | |
|---|---|---|---|---|---|---|
| | | | ML | sZ | cT | 1Q |
| N/A (Zero-Shot) | Mask | 80% | ML | SZ | CT | 1Q |
| | Pattern | 0% | 76492385 | 739482561 | 74928536 | 749238561 |
| Fine-Tuned on Masks | Mask | 94% | ML | sZ | cT | 1Q |
| | Pattern | 0% | R1 | M5 | 4model | D4 |
| Fine-Tuned on Ishihara | Mask | 0% | 7s | 24 | n6 | 36 |
| | Pattern | 0% | tH | vF | 6t | 58 |

## 5.4 ABLATIONS AND ADDITIONAL ANALYSES

**Image Resolution Ablation.** To understand the impact of image resolution, we conducted an ablation study by evaluating model performance across a range of resolutions from 300px to 1300px (Table5). The results were twofold. First, models that performed poorly at the native 900px resolution, such as Qwen VL Max, failed across all tested resolutions, which reinforces the claim that they have a fundamental perceptual deficit. Second, we uncovered a "squinting effect" in the best-performing model, GPT-4.1, whose accuracy peaked at the lowest resolution, increasing from 34% to 49% at 300px. We hypothesize that downsampling acts as a low-pass filter, smoothing the high-frequency "noise" from the dot patterns and making the underlying character shapes more salient to the model.

Table 5: Image resolution ablation with 95% Wilson confidence intervals (N=100).

| Model | 300px | 500px | 700px | 900px | 1100px | 1300px |
|---|---|---|---|---|---|---|
| GPT 4.1 | $49.0 \pm 9.8\%$ | $29.0 \pm 8.9\%$ | $31.0 \pm 9.1\%$ | $34.0 \pm 9.3\%$ | $30.0 \pm 9.0\%$ | $29.2 \pm 8.9\%$ |
| GPT-4.1 mini | $39.0 \pm 9.6\%$ | $18.2 \pm 7.6\%$ | $18.0 \pm 7.6\%$ | $18.0 \pm 7.6\%$ | $10.0 \pm 5.9\%$ | $6.0 \pm 4.6\%$ |
| Qwen VL Max | $1.0 \pm 2.0\%$ | $1.0 \pm 2.0\%$ | $1.0 \pm 2.0\%$ | $1.0 \pm 2.0\%$ | $1.0 \pm 2.0\%$ | $1.0 \pm 2.0\%$ |
| Pixtral Large | $1.0 \pm 2.0\%$ | $0.0 \pm 1.9\%$ | $1.0 \pm 2.0\%$ | $0.0 \pm 1.9\%$ | $0.0 \pm 1.9\%$ | $0.0 \pm 1.9\%$ |

**MLLM Failure Patterns.** A qualitative analysis of MLLM failures on the HueManity benchmark reveals that, unlike humans whose errors are predictable, MLLMs exhibit three distinct and more fundamental failure patterns when their perceptual abilities are overwhelmed (see Appendix E for more details).

- Hallucination: Models often generate completely unrelated and overly complex text, such as turning a two-character string into a full phrase (e.g., 'MUST SEE'). This happens when the perceptual challenge overwhelms their visual processing.

- Evasion: They frequently engage in evasive behavior, either by describing the image as an 'Ishihara test' without attempting to identify the characters or by explicitly stating they are unable to perform the task. This suggests an internal confidence threshold triggers a pre-programmed failure response.

- Erratic Outputs: Their outputs are often erratic and unpredictable, ranging from random strings and nonsensical numbers to flawed, repetitive patterns (e.g., '[L1L1L1]'). This unpredictability points to a lack of robust and stable visual feature extraction.

**Quantitative Correlation with Real-World Benchmarks.** To validate that performance on Hue-Manity translates to real-world capabilities, we correlated model rankings on our alphanumeric pattern task with the human-preference Elo ratings from Vision Arena Chou et al. (2025), a benchmark for diverse, real-world multimodal tasks. Our analysis included the seven models common to both leaderboards: GPT-4.1, GPT-4.1 mini, Claude 3.7 Sonnet, Mistral Medium, Qwen VL Max, Pixtral Large, and LLaVA-v1.6-34B. The analysis revealed a **strong and statistically significant positive correlation** of $\rho = 0.8214\,(p = 0.0227)$. This result provides compelling empirical evidence that HueManity is not an isolated synthetic challenge but effectively measures a foundational perceptual skill that is highly predictive of an MLLM's general performance on complex, real-world tasks. This validates our benchmark as a potent diagnostic tool for identifying critical and generalizable gaps in the core perceptual abilities of current models.

## 6 CONCLUSION AND FUTURE DIRECTIONS

In this work, we introduce HueManity, a large-scale benchmark with 83,850 Ishihara-style images designed to probe the fine-grained visual perception of Multimodal Large Language Models (MLLMs). Our comprehensive evaluation of nine state-of-the-art models reveals a critical weakness: the best-performing MLLM achieved only (33.6%, 3%) accuracies on the numeric and alphanumeric Ishihara pattern recognition tasks, starkly contrasting with the near-perfect scores of human evaluators (99.37%, 93.25%) and a fine-tuned ResNet50 baseline (96.5%, 94.5%). We demonstrate that this perceptual deficit is not easily remedied, as neither in-context learning nor direct fine-tuning yielded improvements, suggesting the failure is rooted in fundamental architectural limitations rather than the task's inherent difficulty. The strong correlation ($\rho = 0.82$) between HueManity performance and rankings on the real-world Vision Arena benchmark validates that our findings are not isolated to a synthetic task but are indicative of a broader, more generalizable weakness in MLLM perceptual abilities.

The deficiencies observed likely stem from vision encoders optimized for high-level semantics at the expense of fine-grained detail, information bottlenecks at the vision-language interface, and pre-training datasets that lack sufficient perceptual challenges. To bridge this gap, future work should focus on developing novel MLLM architectures that preserve low-level visual information, augmenting training datasets with perceptually challenging stimuli, and designing training objectives that explicitly foster robust visual acuity independent of high-level semantic reasoning. We release our dataset and generation code to facilitate further research toward building more perceptually grounded MLLMs.

## 7 LIMITATIONS

While HueManity offers key insights into MLLM perception, we acknowledge its limitations. First, our benchmark centers on a specific task: recognizing alphanumeric characters in Ishihara-style patterns. Although this serves as a strong proxy for discerning patterns in visually cluttered environments, the direct generalization of these findings to the full spectrum of real-world perceptual challenges warrants further study. Second, our dataset primarily explores variations in color and shape. Future work could expand upon our methodology to assess other perceptual dimensions, such as sensitivity to texture, orientation, and motion. Finally, our study presents a snapshot of the rapidly evolving MLLM landscape, future models with different architectures may exhibit different performance characteristics.

ETHICS STATEMENT

The authors have read and adhered to the ICLR Code of Ethics. The HueManity benchmark is procedurally generated and consists of alphanumeric characters and color patterns, it is free from personally identifiable information (PII) and inherent social or demographic biases. The human baseline study involved 16 adult volunteers who participated in a non-invasive visual recognition task. All data collected from participants was fully anonymized to protect their privacy.

REPRODUCIBILITY STATEMENT

We are committed to ensuring the reproducibility of our research. To this end, we will make the complete HueManity dataset (83,850 images) and all source code publicly available upon publication. This includes the code used for the procedural generation of the Ishihara-style stimuli. The methodology for our data creation process is described in detail in Section 3, with further specifics on the principled color pair selection available in Appendix D. All experimental setups, including the specific prompts used for MLLM evaluation, are detailed in Section 4. A comprehensive list of the nine MLLMs evaluated, their versions, and all inference parameters can be found in the Model Inventory in Appendix A (Table 6). The implementation and training details for the ResNet50 baseline and the methodology for the human baseline are described in the Section 4. Details for the fine-tuning and ablation experiments are provided in Section 5. We believe these resources provide the necessary details for the community to fully reproduce our results and build upon this work.

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

## A  MODEL INVENTORY

## B  LABEL SPACES & CHANCE BASELINES

**Character universe.** We define a character set $\mathcal{C}$ used to compose two-character labels. Starting from decimal digits and ASCII letters with case,

$$\{0,\ldots,9,\ A,\ldots,Z,\ a,\ldots,z\} \quad \text{(62 symbols)},$$

we exclude four visually ambiguous characters: lowercase $\ell$ and uppercase $\{I, J, O\}$. Thus,

$$|\mathcal{C}| = 62 - 4 = 58.$$

| Model | Provider | Version | Temp / Top-p | Max Tokens |
|-------|----------|---------|--------------|------------|
| GPT-4.1 mini | OpenAI | gpt-4.1-mini-2025-04-14 | 1.0 / 1.0 | 1024 |
| GPT-4.1 | OpenAI | gpt-4.1-2025-04-14 | 1.0 / 1.0 | 1024 |
| Claude 3.7 Sonnet | Anthropic | claude-3-7-sonnet-20250219 | 1.0 / 1.0 | 1024 |
| LLaVA-v1.6-7B | Ollama | llava:7b | 0.8 / 0.9 | -1 (inf) |
| LLaVA-v1.6-13B | Ollama | llava:13b | 0.8 / 0.9 | -1 (inf) |
| LLaVA-v1.6-34B | Ollama | llava:34b | 0.8 / 0.9 | -1 (inf) |
| Mistral-small3.1-24b | Mistral AI | mistral-small-3.1-24b-instruct | 0.7 / 1.0 | 1024 |
| Qwen VL Max | Qwen (OpenRouter) | qwen/qwen-vl-max | 1.0 / 1.0 | 1024 |
| Pixtral | Mistral AI | pixtral-large-2411 | 0.7 / 1.0 | 1024 |

Table 6: Standardized evaluation settings per model. All models were evaluated between May 6-7, 2025. Ollama models were hosted on a single GPU machine with an NVIDIA A100. These generation values are the model/provider defaults and we stick to these values to avoid introducing noise to our evaluations.

### B.1 Tasks and exact label spaces

**Numeric (two digits).** The numeric label space contains all two-digit numbers *without a leading zero*:

$$\mathcal{N} = \{10, 11, \ldots, 99\}, \qquad |\mathcal{N}| = 90.$$

A uniform random-guess baseline is therefore

$$\Pr[\text{random correct}] = \frac{1}{|\mathcal{N}|} = \frac{1}{90} \approx 1.11\%.$$

**Alphanumeric (two characters).** The alphanumeric label space contains *all ordered pairs* from $\mathcal{C}$:

$$\mathcal{A} = \mathcal{C} \times \mathcal{C}, \qquad |\mathcal{A}| = 58^2 = 3364.$$

A uniform random-guess baseline is therefore

$$\Pr[\text{random correct}] = \frac{1}{|\mathcal{A}|} = \frac{1}{3364} \approx 0.0297\%.$$

### B.2 Scoring policy

We compute case-sensitive exact-match accuracy over $\mathcal{N}$ or $\mathcal{A}$, as applicable. Normalization prior to scoring trims whitespace, removes a possible "Answer: " prefix, extracts the first two alphanumerics. Characters outside $\mathcal{C}$ are considered invalid.

## C A Brief Discussion on CIEDE2000 Color Difference

$$\Delta E_{2000} = \sqrt{\left(\frac{\Delta L'}{K_L S_L}\right)^2 + \left(\frac{\Delta C'}{K_C S_C}\right)^2 + \left(\frac{\Delta H'}{K_H S_H}\right)^2 + R_T \left(\frac{\Delta C'}{K_C S_C}\right)\left(\frac{\Delta H'}{K_H S_H}\right)}$$

where 
- $\Delta L'$ is the corrected lightness difference,
- $\Delta C'$ is the corrected chroma difference,
- $\Delta H'$ is the corrected hue difference,
- $K_L, K_C, K_H$ are parametric factors (typically 1),
- $S_L, S_C, S_H$ are weighting functions for lightness, chroma, and hue,
- $R_T$ is a rotation term accounting for hue-chroma interaction.

(1)

The CIEDE2000 score ($\Delta E_{2000}$, Equation 1) Luo et al. (2001) quantifies the perceived difference between two colors more accurately than prior formulae, especially for subtle variations. It calculates a single value representing the "distance" between colors in the perceptually uniform CIE

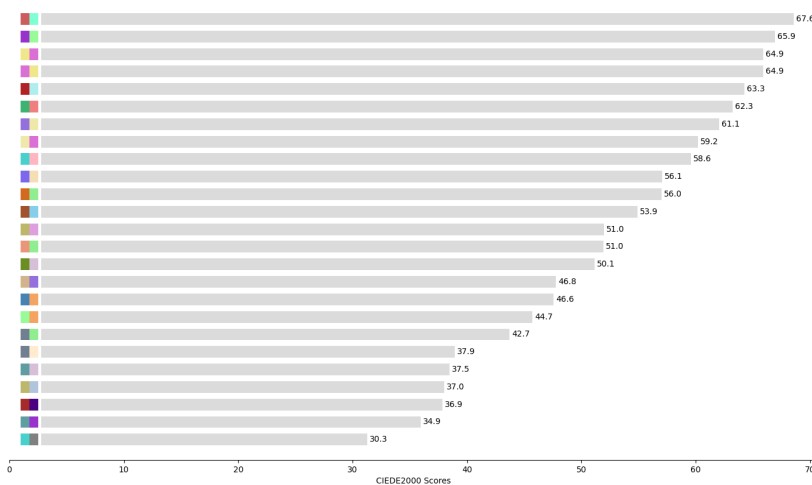

Figure 3: Distribution of CIEDE2000 color difference scores for the 25 selected foreground-background color pairs utilized in the HueManity benchmark.

$L^*a^*b^*$ space, considering lightness, chroma, and hue. In the HueManity benchmark, $\Delta E_{2000}$ was pivotal for systematically designing stimuli. The ability to discern characters in the Ishihara-style plates directly depends on the perceived color contrast between foreground (character) and background dots. This score provided a perceptually relevant, objective method to quantify this contrast, enabling the selection of color pairs across a controlled spectrum of difficulty, refer to Figure 3. This ensured the benchmark could rigorously test visual perception for varying degrees of color discriminability while maintaining stimuli legibility for human comparison, forming a foundational aspect of our dataset's controlled experimental design.

## D    COLOR PAIRS SELECTION

The selection of appropriate color pairs for the foreground (characters) and background dots was a critical phase in the development of HueManity, undertaken with considerable care to ensure a balance between perceptual challenge and unambiguous human legibility. The process involved several stages:

1. **Initial Candidate Generation:** We bootstrapped the process with 15 medium-contrast color pairs generated by LLMs (Gemini, ChatGPT). This initial pool was iteratively refined by evaluating pairs against CIEDE2000 ($\Delta E_{2000}$, Eq. 1) scores and visual checks. We retained promising candidates, modified some, and discarded others, while simultaneously manually crafting and vetting new pairs to meet the benchmark's final requirements (detailed below). This refinement cycle culminated in the selection of 25 distinct pairs for the subsequent validation stages.

2. **Quantitative Contrast Filtering (CIEDE2000):** Each of these candidate pairs then underwent rigorous quantitative analysis using the CIEDE2000 ($\Delta E_{2000}$) color difference formula (Equation 1). This formula is a standard measure in color science, designed to reflect perceptually meaningful differences as perceived by humans. We established a specific target range for the $\Delta E_{2000}$ score, retaining only pairs with contrast values between 25 and 75. The lower bound of 25 was set to ensure sufficient theoretical distinguishability for individuals with normal color vision, preventing pairs that would be inherently ambiguous. The upper bound of 75 aimed to exclude pairs with excessively high contrast, which might render the perceptual task trivial and deviate from the subtle challenge intended.

3. **Balanced Contrast Distribution:** A key objective during selection was to ensure the benchmark included stimuli across a spectrum of difficulty levels related to color similarity. Therefore, we deliberately curated the final set of 25 color pairs to achieve an approximately equal distribution around a $\Delta E_{2000}$ score of 50. This threshold is grounded in

color science literature, often considered a point distinguishing more subtle (scores $< 50$) from more clearly distinct (scores $> 50$) color differences. We aimed for roughly half the selected pairs to fall below this threshold and half above, ensuring HueManity evaluates performance across varying, literature-informed degrees of color contrast difficulty.

4. **Manual Verification and Legibility Check:** Recognizing that a single numerical contrast score like $\Delta E_{2000}$ captures overall perceived difference but may not fully account for the complex interplay of hue, saturation, and luminance components, especially when rendered as dots and subjected to further transformations (gradient, color, and light shifts), a crucial final step of manual verification was performed. It is hard to quantify the nuanced visual impact of these combined factors with a single metric, therefore for every color pair that passed the quantitative filtering, sample HueManity images were generated. These renderings were meticulously inspected by the authors. The primary goal was to reject pairs where the characters, despite an acceptable overall contrast score, appeared visually too similar to the background due to the specific combination of hue, saturation, luminance, or the effect of the applied shifts. This ensured that the embedded alphanumeric characters were clearly legible and that the pattern recognition was unambiguous for human observers with normal color vision. Any pairs that resulted in ambiguous characters or were otherwise problematic during this visual check were discarded.

This multi-stage process, combining LLM-based idea generation, principled quantitative filtering based on color science, a balanced distributional strategy, and crucial human judgment to account for complex visual interactions, resulted in the final curated set of 25 color pairs. This ensures that the stimuli used in HueManity are not only theoretically sound but also practically validated for fairness, legibility, and the intended level of perceptual challenge.

Table 7: The 25 Curated Color Pairs Used in the HueManity Benchmark.

| Foreground Color | Background Color | (Foreground — Background) RGB Values |
|---|---|---|
| Firebrick | PaleTurquoise | (178, 34, 34) — (175, 238, 238) |
| Sienna | SkyBlue | (160, 82, 45) — (135, 206, 235) |
| OliveDrab | Thistle | (107, 142, 35) — (216, 191, 216) |
| DarkOrchid | PaleGreen | (153, 50, 204) — (152, 251, 152) |
| SteelBlue | SandyBrown | (70, 130, 180) — (244, 164, 96) |
| MediumSeaGreen | LightCoral | (60, 179, 113) — (240, 128, 128) |
| Chocolate | LightGreen | (210, 105, 30) — (144, 238, 144) |
| Khaki | Orchid | (240, 230, 140) — (218, 112, 214) |
| DarkKhaki | Plum | (189, 183, 107) — (221, 160, 221) |
| Orchid | Khaki | (218, 112, 214) — (240, 230, 140) |
| IndianRed | Aquamarine | (205, 92, 92) — (127, 255, 212) |
| PaleGreen | SandyBrown | (152, 251, 152) — (244, 164, 96) |
| Brown | Indigo | (165, 42, 42) — (75, 0, 130) |
| CadetBlue | DarkOrchid | (95, 158, 160) — (153, 50, 204) |
| PaleGoldenrod | Orchid | (238, 232, 170) — (218, 112, 214) |
| MediumTurquoise | Grey | (72, 209, 204) — (128, 128, 128) |
| SlateGrey | LightGreen | (112, 128, 144) — (144, 238, 144) |
| MediumSlateBlue | Wheat | (123, 104, 238) — (245, 222, 179) |
| MediumTurquoise | LightPink | (72, 209, 204) — (255, 182, 193) |
| DarkSalmon | LightGreen | (233, 150, 122) — (144, 238, 144) |
| Tan | MediumPurple | (210, 180, 140) — (147, 112, 219) |
| MediumPurple | PaleGoldenrod | (147, 112, 219) — (238, 232, 170) |
| SlateGray | BlanchedAlmond | (112, 128, 144) — (255, 235, 205) |
| CadetBlue | Thistle | (95, 158, 160) — (216, 191, 216) |
| DarkKhaki | LightSteelBlue | (189, 183, 107) — (176, 196, 222) |

# E    QUALITATIVE ANALYSIS OF MLLM FAILURE PATTERNS

This section details common failure patterns observed in Multimodal Large Language Models (MLLMs) when tasked with identifying alphanumeric characters embedded in Ishihara-style dot pat-

terns from the HueManity dataset. These observations stem from a comparative analysis of MLLM responses against human performance and ground truth data. Notably, human visual perception proved highly accurate on these tasks, with any infrequent errors typically involving confusion between graphically similar characters. In contrast, MLLMs exhibited distinct and more fundamental failure modes.

### E.1 PREVALENT HALLUCINATION OF UNRELATED OR OVERLY COMPLEX CHARACTERS

A dominant failure mode across multiple MLLMs was the generation of characters, words, or even entire phrases that bore no resemblance to the two-character ground truth. This phenomenon of "hallucination" often resulted in outputs significantly more complex or contextually incongruous than the target stimuli. For instance, in the alphanumeric task, a model such as `Claude 3.7 Sonnet` might interpret a simple two-letter combination as a short phrase (e.g., responding with "MUST SEE" or "SOLU" for simple targets like "Rw" or "Tv"). Similarly, `llava-7b` could produce non-sensical strings like "HQJHSTOS", and `LLaVA-13b` occasionally generated contextually unrelated phrases like "[G3T1NGST4RT3D]". The numeric task was not immune — for a two-digit number, `Claude 3.7 Sonnet` was observed to list a sequence of unrelated two-digit numbers. This pattern suggests that when the fine-grained perceptual challenge overwhelms the MLLMs' visual processing, they may default to generating text that, while perhaps linguistically plausible, is detached from the actual visual content.

### E.2 FREQUENT RESORT TO DESCRIPTIVE EVASION OR EXPLICIT ADMISSION OF INABILITY

Rather than consistently attempting to identify the embedded characters, many MLLMs frequently defaulted to one of two evasive strategies: providing a general description of the image (often correctly identifying it as a color vision test) or explicitly stating their incapacity to discern any characters. This behavior contrasted significantly with human participants, who invariably attempted the identification task. For example, models like `GPT-4.1 Mini` and `Mistral-small3.1-24b`, when presented with alphanumeric stimuli, often responded by describing the image as an Ishihara test but stated they could not clearly identify specific characters. In the numeric task, `Claude 3.7 Sonnet` sometimes offered similar descriptive evasions, asserting no number was visible and describing the circular dot pattern. Furthermore, some models, such as `LLaVA-34b`, occasionally provided categorical statements of inability, indicating they could not recognize or interpret images and requesting a description or textual input instead. This pattern suggests that MLLMs may possess internal confidence thresholds that, when triggered by low-confidence visual parsing, lead to evasive or pre-programmed "unable to process" responses rather than a forced, best-guess attempt at character recognition.

### E.3 ERRATIC, UNPREDICTABLE, AND SYSTEMATICALLY FLAWED OUTPUT PATTERNS

MLLM outputs were frequently characterized by their erratic and unpredictable nature. This included the generation of seemingly random strings of characters, peculiar systematic but incorrect patterns, or extreme numerical inventions far removed from the two-character target. This high variance in error types was observed both across different models for the same input and within the outputs of a single model across different images. For instance, when presented with the same alphanumeric target (e.g., "Wh"), while one model (`GPT-4.1`) might respond almost correctly, others exhibited diverse failures: `Claude 3.7 Sonnet` produced an unrelated number ("4726"), `LLaVA-13b` generated an exceptionally long string of sequential numbers, and `Qwen VL Max` incorrectly reasoned the presence of a different number ("12"). Some incorrect outputs also suggested flawed systematic processing, such as `LLaVA-13b` responding with a patterned string like "[L1L1L1]" for one target or generating extremely long, patterned numeric strings for others. Lengthy, seemingly gibberish character strings were also common from models like `LLaVA-7b`. This unpredictability underscores a lack of robust and stable visual feature extraction and interpretation, contrasting with human visual processing, which tends towards predictable errors based on similarity.

## F   USAGE OF GENERATIVE AI TOOLS

We utilized Generative AI tools to help improve the language, phrasing, and readability of this manuscript.

