# OpenReview forum: "HueManity: Probing Fine-Grained Visual Perception in MLLMs"
_ICLR.cc/2026/Conference — ICLR 2026 Conference Withdrawn Submission_

### Official Review · Reviewer_zQip · 2025-10-31

**Soundness:** 2
**Presentation:** 3
**Contribution:** 2
**Rating:** 4
**Confidence:** 4

**Summary:**

This paper proposes a synthetic visual text-reading benchmark to study how well MLLMs can perceive alphanumeric patterns in an uncommon visual representation. Specifically, the proposed benchmark represents two-letter texts in an image as a set of circles of one color against a background filled by circles of another color, known as an Ishihara Pattern. The paper shows that several popular MLLMs perform poorly in reading the text from such images, whereas human and ResNet classifiers perform almost perfectly.

**Strengths:**

The paper is well-written and easy to follow. The benchmark is novel and does reflect a striking limitation in MLLMs’ visual perception, which pushes against the misconception that MLLMs’ can outperform humans in all simple visual tasks. The paper also considered several MLLMs, both commercial and open-source, which increases its value as a benchmark for future MLLM development.

**Weaknesses:**

1. The paper misses several related papers that study the ability of MLLMs on perceiving visual details [1, 2, 3, 4], and thus does not properly place its findings in the context of other existing evidence to clarify novelty and relevance.

2. Text recognition datasets (eg, TextVQA) measure the same capability that this paper tries to measure: how well can MLLMs read text in various visual settings. Given that TextVQA contains extensive variations of text and background in real world settings, it can provide a more reliable measure of MLLMs’ overall text reading capability compared to synthetic data which may cause distribution shift. This makes the practical utility of the proposed benchmark a bit unclear: if a model L1 outperforms another model L2 on HueManity, what does it mean for real-world applications? Consider this in comparison to TextVQA that has a clear connection to real-world applications.

3. The paper only considers a single prompt and does not explore the effect of prompt variations on the performance. For example, does including instructions such as “There is a letter on an Ishihara Pattern in the image…” and/or removing the exclusion instructions, help improve the performance? This is important because MLLMs performance is quite sensitive to the prompt.

4. The paper does not explore the causes for the discovered difficulty. The mentioned potential causes in Lines 348-359 are speculations without any evidence. Quantitatively exploring some of these speculative causes can strengthen the paper’s contributions.

5. The results of MLLM fine-tuning seems to contradict the ResNet training performance. The paper does point out that it is very surprising that fine-tuning MLLMs on the task does not result in improvements, but does not explore this surprising observation further. For example, is this because the LLM is also finetuned instead of just its vision encoder? Is this because of bad choices of hyperparameters when fine-tuning? There are many missing details here that make the results unreliable. It is also unclear whether this is just a problem with Gemma, or the same applies to other MLLMs.

6. Providing quantitative results for the “MLLM Failure Patterns” could substantiate the claims in lines-425-439.

7. In Tables 2-5, Wilson intervals seem incorrect since they should not be symmetric and fall outside of [0,1]. Reporting the actual confidence interval bounds instead of +- will clarify this.

[1] MLLMs Know Where to Look: Training-Free Perception of Small Visual Details with Multimodal LLMs. ICLR 2025.

[2] Understanding Depth and Height Perception in Large Visual-Language Models. CVPR 2025.

[3] V*: Guided Visual Search as A Core Mechanism in Multimodal LLMs. CVPR 2024.

[4] Exploring Perceptual Limitation of Multimodal Large Language Models. 2024.

**Questions:**

1. Can the authors explain/clarify why fine-tuning the MLLM does not improve its performance? This seems contradictory to a lot of prior research and the fine-tuning results of ResNest. Also, does this happen to other MLLMs besides Gemma?

2. Can different prompts (eg, explicitly mentioning the Ishihara Pattern) change the performance?

---

> ### Author Response · Authors · 2025-11-19
> **Reply to Reviewer zQip (1/3)**
>
> We thank the reviewer for highlighting the novelty of our benchmark and for recognizing that it challenges the misconception of MLLM perceptual superiority. Below we address the weaknesses mentioned:
>
> ---
>
> **1. Missing Related Work:** We appreciate these references and will incorporate [1-4] into our related work section, providing a detailed comparison of our work against them in the final revision.
>
> ---
>
> **2. Comparison to TextVQA:** While TextVQA is excellent for "in-context" reading, it often relies on semantic priors (e.g., a menu layout suggests food items). HueManity tests the raw perception capability of the model without these semantic clues, beyond pure “text-reading” capabilities. Additionally, our procedural generation ensures that the data is unseen by the models and is not exposed in its training.
>
> To validate real-world relevance, we highlight the strong correlation with the Vision Arena (ρ=0.82, p=0.0227) benchmark in Section 5.4. Vision Arena benchmark consists of diverse real-world tasks and also contains the TextVQA dataset, and is therefore, indirectly included in our comparison with Vision Arena benchmark. This correlation suggests that HueManity is a critical indicator of general performance, if a model L1 outperforms L2 on HueManity, it will likely perform better than L2 in the real-world as well.
>
> ---
>
> **3. Prompt Sensitivity:** We agree with the reviewer’s comments about prompt sensitivity. While building this benchmark, we had iterated over several variants of prompts. We note that performance of the MLLMs we evaluated did not change significantly by varying the prompts. In many cases, the current prompt gave best results compared to others. We are currently working on reproducing these results and will share them as soon as possible.
>
> ---
>
> **4. Causes for Discovered Difficulty:** We draw attention to Appendix E, where we qualitatively analyze different failure patterns. A deep dive into the causes of MLLM failures is outside the scope of our current work as it may require modifications to model architectures and we leave it for future work. We share this benchmark in the hope that the community can diagnose similar issues and work on fixing this issue with perception abilities using our benchmark.

---

> > ### Author Response · Authors · 2025-11-19
> > **Reply to Reviewer zQip (2/3)**
> >
> > **5. Fine-Tuning Contradiction:** We appreciate the reviewer asking this question and believe the contrast between the ResNet and MLLM fine-tuning results is one of our most important findings. The ResNet (CNN) succeeds because it preserves spatial hierarchies. The MLLM (ViT-based) fails because the encoder likely "smooths out" the high-frequency dot patterns into texture tokens, destroying the character information before it reaches the LLM.
> >
> > This is supported by our ablation study showing that lowering image resolution actually improved GPT-4.1's performance, suggesting the high-fidelity signal is being treated as noise.
> >
> > Moreover, we validated our training pipeline by fine-tuning Gemma-3-4B on the Text Mask task. The model successfully improved from 80% to 94% accuracy. This confirms that our hyperparameters were correct and that the failure on Ishihara plates is due to architectural limitations in perceiving noisy patterns, and not an optimization issue.
> >
> > For verification of fine-tuning setup, we provide the following response (as also shared with reviewer GXxi):
> > * Verifying Implementation Issues: We appreciate the question on the training setup. To ensure our pipeline was valid, we conducted a control experiment detailed in Section 5.3. Using the exact same setup (500 samples, 3 epochs), we fine-tuned the model on the Text Mask task, where accuracy successfully improved from 80% to 94%. We also observed the training loss plateauing within this setup, indicating convergence.
> > * Verifying hyperparameters and overfitting: We also varied the hyperparameters such as epochs, number of examples, etc. but the train accuracy does not decrease, even though we see the training loss decrease and then plateaus. This shows that the model is not overfitting on the dataset. The model learns the output format (predicting 2 chars) but fails to learn any generalizable perception skills for the task.
> >
> > ---
> >
> > **6. Incorrect Wilson Intervals:** We thank the reviewer for pointing this out. Instead of the symmetric “±” format, we will report the exact lower and upper bounds of the Wilson score intervals.

---

> > > ### Author Response · Authors · 2025-11-19
> > > **Reply to Reviewer zQip (3/3)**
> > >
> > > **7.  Quantitative results for MLLM failure patterns:** We thank the reviewer for this suggestion. While we provide qualitative examples for the MLLM behavior patterns in Appendix E, we present the following quantitative analysis for the common failure modes. We evaluated 100 samples for number and alphanumeric tasks each.
> > >
> > > | Model                    | Task          | Total Entries | Correct Count | Evasion Count | Hallucination Count | Erratic Outputs Count |
> > > |--------------------------|---------------|---------------|----------------|----------------|-----------------------|------------------------|
> > > | GPT-4.1                  | Number        | 100           | 2              | 0              | 0                     | 98                     |
> > > | GPT-4.1 Mini             | Number        | 100           | 1              | 0              | 0                     | 99                     |
> > > | Claude 3.7 Sonnet        | Number        | 100           | 0              | 0              | 3                     | 97                     |
> > > | llava-7b                 | Number        | 100           | 0              | 10             | 55                    | 35                     |
> > > | llava-13b                | Number        | 100           | 0              | 15             | 18                    | 67                     |
> > > | llava-34b                | Number        | 100           | 1              | 2              | 5                     | 92                     |
> > > | mistral-small-3.1-24b    | Number        | 100           | 0              | 61             | 12                    | 27                     |
> > > | Pixtral                  | Number        | 100           | 0              | 0              | 0                     | 100                    |
> > > | Qwen VL Max              | Number        | 100           | 0              | 31             | 3                     | 66                     |
> > > | GPT-4.1                  | Alphanumeric  | 100           | 0              | 0              | 0                     | 100                    |
> > > | GPT-4.1 Mini             | Alphanumeric  | 100           | 0              | 0              | 1                     | 99                     |
> > > | Claude 3.7 Sonnet        | Alphanumeric  | 100           | 0              | 90             | 0                     | 10                     |
> > > | llava-7b                 | Alphanumeric  | 100           | 0              | 20             | 74                    | 6                      |
> > > | llava-13b                | Alphanumeric  | 100           | 0              | 3              | 20                    | 77                     |
> > > | llava-34b                | Alphanumeric  | 100           | 0              | 14             | 28                    | 58                     |
> > > | mistral-small-3.1-24b    | Alphanumeric  | 100           | 0              | 21             | 42                    | 37                     |
> > > | Pixtral                  | Alphanumeric  | 100           | 0              | 63             | 36                    | 1                      |
> > > | Qwen VL Max              | Alphanumeric  | 100           | 0              | 11             | 24                    | 65                     |
> > >
> > > ---

---

### Official Review · Reviewer_Liuz · 2025-10-31

**Soundness:** 2
**Presentation:** 2
**Contribution:** 2
**Rating:** 4
**Confidence:** 2

**Summary:**

This work proposes HueManity, a new benchmark designed to evaluate the fine-grained visual perception capabilities of multimodal large language models (MLLMs). The benchmark contains 83,850 samples, each consisting of a "blind-test" image (similar to Ishihara plates) and a corresponding question that requires the model to recognize embedded letters or numbers. This task is highly accessible for humans, achieving over 99% accuracy, and is also easily handled by lightweight models such as fine-tuned ResNet-50. In contrast, state-of-the-art MLLMs struggle significantly on this task, revealing a notable gap in their ability to perform precise, low-level visual perception—even for seemingly simple recognition tasks.

**Strengths:**

* This paper identifies a critical deficiency in modern MLLMs: their surprisingly weak performance in fine-grained visual perception, despite strong performance on higher-level vision-language tasks.

* The proposed benchmark, HueManity, is well-designed and presents a valuable resource for the community. It can be widely used in future work to evaluate and diagnose the fine-grained visual understanding capabilities of MLLMs.

* The authors conduct comprehensive experiments demonstrating that even state-of-the-art MLLMs struggle significantly on this task, highlighting the challenge of achieving robust, low-level perceptual accuracy in current multimodal models.

**Weaknesses:**

* This work focuses on a single aspect of visual understanding—recognizing characters in color-patterned images—which is relatively narrow compared to existing MLLM benchmarks. Modern benchmarks typically evaluate multiple capabilities, including low-level perception, high-level reasoning, OCR, and knowledge integration. While this task presents a challenging variant of OCR, the scope of the benchmark is limited in covering the broader spectrum of multimodal understanding expected from MLLMs.

* The benchmark appears to have high redundancy. Given the simple and repetitive nature of the task—overlaying letters and numbers on textured or colorful backgrounds—it is questionable whether 83,850 samples are necessary to reliably evaluate current MLLMs. A much smaller set (e.g., a few thousand examples) might suffice for stable assessment, especially considering that the variation is primarily in color and noise patterns rather than semantic complexity.

* While framed as an MLLM capability, the core task primarily tests the vision encoder’s ability to extract fine-grained visual features under visual noise. The language component is minimal (simple recognition questions), suggesting that the bottleneck lies in the visual encoder rather than the multimodal reasoning or language generation pipeline. Therefore, the focus may be better positioned as evaluating the fine-grained perception capabilities of vision encoders, rather than MLLMs as a whole.

**Questions:**

Please see weaknesses.

---

> ### Author Response · Authors · 2025-11-19
> **Reply to Reviewer Liuz**
>
> We thank the reviewer for identifying our benchmark as a valuable and well-designed resource for the community.
>
> ---
>
> **1. Scope and Narrowness of the Task:**
> Most of the popular benchmarks for MLLMs evaluate perception abilities in combination with reasoning abilities of MLLMs.
>
> We acknowledge that recognizing Ishihara patterns is a specific task, but we view this specificity as a strength. By stripping away high-level reasoning, HueManity acts as a "unit test" for the visual recognition. If a model cannot perform figure-ground segregation, its success on broader benchmarks may be brittle. This narrow focus allows us to isolate the exact failure point-perceptual grouping - which broad benchmarks can obscure.
>
> Instead of trying to find more gaps in connection with their reasoning abilities and more cases of hallucinations, we attempt to showcase and benchmark a fundamental flaw in the design of vision encoders for high-performing MLLMs.
>
> ---
>
> **2. Dataset Size and Redundancy (83,850 samples):**
> We agree that 83k samples are not needed for evaluating the MLLMs. For all our evaluations, we used a much smaller subset of 1000 images and found it to be insightful. We provide the complete dataset of 83,850 samples for two key reasons. Completeness: We generated all possible character combinations to eliminate combinatorial bias and ensure statistical coverage across all color pairs. Training Potential: This enables future research to attempt large-scale pre-training or robust fine-tuning experiments that require data magnitude.
>
> ---
>
> **3. Focus on Vision Encoder vs. MLLM:**
> We agree that the bottleneck likely lies in the vision encoder. However, since the most widely used MLLMs (e.g., GPT-4, Claude) are closed systems where the encoder cannot be evaluated in isolation, benchmarking the system as a whole is the only viable diagnostic method.
>
> Modern MLLMs are promoted as general-purpose foundation models expected to handle fine-grained visual understanding. Our results reveal a critical gap: while they are shown to exhibit strong reasoning in some benchmarks, they struggle with visually grounded perception tasks that should fall within their capabilities.
>
> Our findings - such as the "Squinting Effect" where lower resolution improves performance - inform how future encoders should be designed to preserve high-frequency information.
>
> In this sense, the benchmark serves as a necessary atomic step: by isolating and evaluating fine-grained visual perception failures, we can identify critical bottlenecks that impede holistic multimodal understanding—a central goal of MLLMs.

---

> > ### Comment · Reviewer_Liuz · 2025-11-27
> >
> > Thanks for the responses. I will take it into account when makeing the final decision.

---

### Official Review · Reviewer_GXxi · 2025-10-31

**Soundness:** 2
**Presentation:** 3
**Contribution:** 2
**Rating:** 2
**Confidence:** 4

**Summary:**

This paper introduces HueManity, a new benchmark for assessing fine-grained visual perception in Multimodal Large Language Models (MLLMs). It contains 83,850 Ishihara-style images designed to evaluate the fine-grained perceptual ability of MLLMs. The benchmark embeds alphanumeric characters within colored dot patterns to test whether models can recognize subtle visual patterns. The evaluation reveals that even top-performing models such as GPT-4.1, Claude 3.7 Sonnet, Qwen-VL Max, LLaVA-v1.6, and Pixtral perform poorly compared to human participants and a fine-tuned ResNet-50 baseline. The authors claim that these results expose a critical weakness in the perceptual grounding of MLLMs.

**Strengths:**

1.	The paper is well written and clearly structured, making it easy to follow.
2.	It evaluates several state-of-the-art MLLMs, including GPT-4.1, Claude 3.7 Sonnet, Qwen-VL Max, LLaVA-v1.6, and Pixtral, across two tasks: the Number Recognition Task and the Alphanumeric Recognition Task.
3.	The work provides a comparative analysis with existing MLLM benchmarks. However, some key benchmarks (e.g., MMVP [1],  MERLIM [2] and MME [3]) are missing from the evaluation.

[1] Eyes Wide Shut? Exploring the Visual Shortcomings of Multimodal LLMs (Tong et al., CVPR 2024)
[2] MERLIM: Multi Modal Evaluation Benchmark for IT LVLMs (Villa et al., CVPRW 2025)
[3] MME: A Comprehensive Evaluation Benchmark for Multimodal Large Language Models (Fu et al., 2023).

**Weaknesses:**

1.	The paper mainly reports a failure case of existing models but offers no new theoretical insights. Prior work such as Eyes Wide Shut [1] and MERLIM [2] has already shown that the visual backbones of MLLMs fail to capture fine-grained visual details.
2.	HueManity measures only color-based figure–ground discrimination under a single visual structure (Ishihara-style dots). While the idea is well motivated, it represents only a narrow and somewhat artificial subset of visual examples for evaluating  fine-grained visual perception. Other state-of-the-art benchmarks (e.g., MMVP [1], MERLIM [2], MME [3] with its OCR tasks) address this challenge from a more realistic perspective.
3.	The LoRA fine-tuning on Gemma-3-4B using only 500 samples and 3 epochs is too limited to support the claim that the issue is unlearnable. The results likely stem from optimization instability (overfitting), data non-representativeness, or implementation issues rather than a fundamental perceptual incapacity.
4.	Although the paper lists three contributions, the third one overlaps with and is effectively part of the first.

**Questions:**

1.	You claim that fine-grained perception is crucial for MLLMs, but is recognizing Ishihara-style digits truly representative of real-world perceptual challenges? What new insights does this benchmark provide beyond state-of-the-art alternatives such as MMVP [1], MERLIM [2], or MME [3] (which already include OCR-style tasks)?
2.	How did you ensure that the alphanumeric strings are balanced across color pairs and not biased by particular hues or contrast levels?

---

> ### Author Response · Authors · 2025-11-19
> **Reply to Reviewer GXxi**
>
> We thank the reviewer for mentioning that the paper is well-written and clearly structured, and for appreciating our comprehensive evaluation of state-of-the-art models. Below, we address the weaknesses mentioned:
>
> ---
>
> **1. Novel theoretical insights compared to existing benchmarks:**
>
> We agree that these benchmarks provide valuable insights; however, they evaluate perception entangled with reasoning.
> Specifically, papers like “Eyes Wide Shut” include questions focussing on semantic and abstract concepts like “left or right”, “floor”, “yellow animal”, which allow models to rely on language priors. Similarly, MERLIM also evaluates hallucinations in presence or absence of objects in the datasets and questions semantic understanding of the image. These other datasets also have a much higher performance of these models on them. For example: MME OCR has almost 95% performance on GPT-4V which is a whole generation behind the models that we evaluate of Huemanity and get single digit performances. Similarly, GPT-4V has ~40% performance on MMVP.
>
> HueManity is unique because it disentangles perception from semantics. There is no context to guess the number '26'—the model must perceive the dots and can not rely on semantic reasoning to predict the answer.
>
> ---
>
> **2. Realistic perspective compared to other benchmarks:**
>
> Regarding real-world relevance, we point to our analysis in Section 5.4, which reveals a strong, statistically significant correlation (ρ=0.82, p=0.0227) between HueManity performance and the Vision Arena leaderboard. Vision Arena is a robust benchmark with diverse real-world tasks that are evaluated by human judges. This indicates that HueManity is a reliable predictor of robustness in the wild and failure here suggests likely failure on real-world tasks involving clutter or low contrast. Our use of synthetic data ensures scalability and zero contamination, preventing models from solving the task via memorization of internet data. Furthermore, the high performance on the mask detection control task confirms this is not a simple OCR failure, but a specific deficit in fine-grained perception.
>
> ---
>
> **3. Validity of Fine-Tuning:**
>
> * Verifying Implementation Issues: We appreciate the question on the training setup. To ensure our pipeline was valid, we conducted a control experiment detailed in Section 5.3. Using the exact same setup (500 samples, 3 epochs), we fine-tuned the model on the Text Mask task, where accuracy successfully improved from 80% to 94%. We also observed the training loss plateauing within this setup, indicating convergence.
> * Verifying hyperparameters and overfitting: We also varied the hyperparameters such as epochs, number of examples, etc. but the train accuracy does not decrease, even though we see the training loss decrease and then plateaus. This shows that the model is not overfitting on the dataset. The model learns the output format (predicting 2 chars) but fails to learn any generalizable perception skills for the task.
> * Regarding data non-representativeness: We generated the complete set of Ishihara-style patterns for our task, and then chose the train-test splits randomly. Any examples in the training set are representative and in-distribution with the test set.
> All these aspects point to the idea that the perceptual capacity of the MLLMs are limited to successfully learn this task.
>
> ---
>
> **4. Question about Data Balance:**
>
> We ensured balanced generation by addressing both color and content biases. We selected the color pairs based on the CIEDE2000 (ΔE2000​) contrast score and curated the final set for a balanced distribution of contrast values. This rigorous selection was reinforced by manual verification to account for complex hue and luminance interactions. Additionally, we eliminated string selection bias by generating the complete set of all possible two-character alphanumeric combinations. We refer to Appendix D for the detailed methodology.

---

### Official Review · Reviewer_nA9E · 2025-10-31

**Soundness:** 3
**Presentation:** 3
**Contribution:** 2
**Rating:** 4
**Confidence:** 3

**Summary:**

The paper uncovers a striking failure mode: mainstream MLLMs struggle to classify Ishihara-style images—even though they excel on general vision benchmarks—whereas humans and a fine-tuned ResNet-50 perform almost flawlessly. To quantify and mitigate this gap, the authors automatically generate a large-scale Ishihara-style dataset (83 850 images) and benchmark leading MLLMs. They hope the corpus will serve as a safety probe and spur progress on “out-of-distribution” visual reasoning.

**Strengths:**

This paper reveals an intriguing blind spot of current MLLMs and highlights a new axis for robustness research; the observation may catalyze broader studies on rare or specially structured imagery.

**Weaknesses:**

This paper does not investigate whether lightweight fine-tuning (rather than mere in-context learning) can already lift MLLM accuracy to near-human levels. If the deficit can be erased with a few gradient steps, the issue—and the accompanying dataset—may merit only limited attention.

**Questions:**

No

---

> ### Author Response · Authors · 2025-11-19
> **Reply to Reviewer nA9E**
>
> We thank the reviewer for recognizing that our work uncovers a striking failure mode and has the potential to catalyze broader robustness research.
>
> Regarding the potential of lightweight fine-tuning: We appreciate this insightful question. As noted in Section 5.3, we performed lightweight fine-tuning (LoRA) experiments on Gemma-3-4B, and the results revealed a critical architectural limitation rather than a simple optimization gap.
>
> * Control Task Success: First, when we fine-tuned the model on the control mask task (text images without the Ishihara pattern), the accuracy improved significantly from 80% to 94%. This confirms our fine-tuning setup was correct and the model is capable of learning when the visual signal is clear.
> * Perceptual Failure: Second, when fine-tuned on the Ishihara task, the accuracy remained at 0%. Instead of learning to perceive the characters, the model learned to hallucinate plausible 2-character strings (e.g., "4G"), ignoring the visual input. Consequently, this fine-tuning led to catastrophic forgetting and destroyed the model's ability to solve the simple mask task, causing accuracy to plummet from 80% to 0%.
>
> These experiments confirm that the issue is not data scarcity, but a fundamental architectural blindness in the MLLM's pre-trained vision encoder.

---

### Note · Authors · 2025-12-04

I have read and agree with the venue's withdrawal policy on behalf of myself and my co-authors.